# Hypodiploid B-Lymphoblastic Leukemia Presenting as an Isolated Orbital Mass Prior to Systemic Involvement: A Case Report and Review of the Literature

**DOI:** 10.3390/diagnostics11010025

**Published:** 2020-12-25

**Authors:** Linyan Wang, Davin C. Ashraf, Benyam Kinde, Robert S. Ohgami, Jyoti Kumar, Robert C. Kersten

**Affiliations:** 1Department of Ophthalmology, The Second Affiliated Hospital, Zhejiang University School of Medicine, Hangzhou 310002, China; wanglinyan@zju.edu.cn; 2Department of Ophthalmology, University of California San Francisco, San Francisco, CA 94143, USA; Davin.Ashraf@ucsf.edu (D.C.A.); Benyam.Kinde@ucsf.edu (B.K.); 3Department of Pathology, University of California, San Francisco, CA 94143, USA; Robert.Ohgami@ucsf.edu; 4Department of Pathology, Stanford University, Stanford, CA 94305, USA; kumarj@stanford.edu

**Keywords:** acute lymphoblastic leukemia (ALL), pre-B cell ALL, orbital neoplasm, extramedullary manifestation

## Abstract

We describe a 4-year-old boy who presented with progressive right periorbital edema and proptosis, with no systemic symptoms, who was found to have B-lymphoblastic leukemia (B-ALL). Magnetic resonance imaging (MRI) showed an enhancing mass centered in the right superolateral extraconal orbit. Orbital biopsy was consistent with B-ALL (CD99, TdT, LCA cocktail, CD34, CD79, CD10, PAX5, MIB1 positive; CD3, CD20 negative). A subsequent bone marrow aspirate confirmed a diagnosis of B-ALL with 80% blasts by flow cytometry and haploid cytogenetic findings. The patient improved clinically after chemotherapy. There are seven cases previously reported in the literature with hematogenous orbital masses at initial presentation of childhood ALL, but all with systemic symptoms or an abnormal complete blood count (CBC) at presentation. Our case is the first report in which an orbital mass preceded detectable systemic or laboratory evidence of ALL. This patient highlights the importance of differentiating benign causes of eyelid swelling from malignant ones.

## 1. Introduction

According to the American Cancer Society, there have been over 6150 new cases of acute lymphoblastic leukemia (ALL) diagnosed in 2020 in the United States, resulting in 1520 deaths. Half of ALL cases occur in children and teenagers. In the United States, ALL is the most common childhood cancer and the leading cause of death from cancer in patients below 20 years of age [1]. ALL accounts for 75% to 80% of childhood leukemias. Despite cure rates exceeding 90% in children, it remains an important cause of morbidity and mortality [2,3]. Orbital and ocular lesions are the third most frequent extramedullary locations of acute leukemia after the meninges and testicles. There are only seven reports of orbital involvement in ALL at initial diagnosis [4]. In a large series of 288 patients with leukemia (including adults and children), ocular involvement was seen in a total of 29.2% of all patients, and in 16.5% of children. Orbital infiltration was seen in only three patients with ALL, all of which occurred after initial diagnosis [5]. Orbital involvement was an indicator of worse prognosis in both acute myeloid leukemia (AML) and ALL [4]. Here, we report the clinical, histopathologic, and genetic findings, as well as management of a 4-year-old boy presenting with an isolated orbital tumor subsequently identified as B-ALL. 

## 2. Case Presentation

This study was conducted in accordance with the Declaration of Helsinki and was exempted from Institutional Review Board (IRB) review at the University of California, San Francisco (UCSF). Written informed consent was obtained from the guardian of the patient.

A 4-year-old Caucasian boy was referred with a 5-week history of progressive right periorbital edema (Figure 1A). His pediatrician initially managed the edema with warm compresses and antibiotic ointment to address a suspected chalazion. After a lack of clinical improvement, the patient was referred to the Department of Oculofacial Plastic Surgery for further evaluation. He was found to have periorbital edema associated with a nontender, palpable medial eyelid mass along the boney orbital rim that was resulting in hypoglobus and markedly restricted upgaze of his right eye. The remainder of this ophthalmic examination was unremarkable. 

He had no systemic symptoms at that time, and labs, including complete blood count (CBC), were normal (white blood cell count, 7.8 × 10^9^/L). Computerized tomography (CT) (Figure 1B) and magnetic resonance imaging (MRI) (Figure 1C) of the brain and orbits demonstrated a 3 cm by 3 cm by 1 cm mass of the right orbit, suspicious for hematologic malignancy or rhabdomyosarcoma. 

Considering the isolated abnormality of orbital mass without any systemic symptoms or signs, the patient underwent an urgent right orbitotomy and biopsy due to concern for rhabdomyosarcoma. Histologic sections of the right orbital mass biopsy instead demonstrated sheets of intermediate-sized immature mononuclear cells with round to irregular nuclear contours, finely dispersed chromatin, inconspicuous nucleoli, and scant cytoplasm (Figure 2A). By immunohistochemistry, the cells were positive for CD99, TdT, LCA cocktail, CD34, CD79a (patchy variable), CD10, and PAX5 (Figure 2B–H). A Ki-67 nuclear labeling index was approximately 70–80%, demonstrating a high proliferation index (Figure 2I). The tumor cells were negative for CD3 and CD20. The overall findings supported a diagnosis of B-lymphoblastic leukemia/lymphoma.

A positron emission tomographic (PET/CT) scan was obtained and did not identify other loci of disease. Subsequent laboratory testing a week after biopsy revealed circulating peripheral blasts 1.89 × 10^9^/L, comprising 14% of circulating white blood cells. He was also found to have a mild microcytic anemia (hemoglobin, 10.5 g/dL) and leukocytosis (white blood cell count, 12.7 × 10^9^/L). A review of the peripheral blood smears (Figure 3A) and bone marrow aspirate smears (Figure 3B) demonstrated numerous circulating small to medium-sized blasts with high nuclear to cytoplasmic ratio, irregular nuclei, open chromatin, small nucleoli, and scant cytoplasm. There were decreased residual hematopoietic precursors. The bone marrow core biopsy was packed (>95% cellularity) with a diffuse infiltrate of blasts (Figure 3C). Flow cytometric studies detected a population of blasts (~80% of all cells) positive for TdT, CD34, CD10, CD19, CD22 (surface and cytoplasmic), CD79a, CD38 (variable), CD123 (variable), and HLA-DR (weak); they were negative for CD3 (surface or cytoplasmic), CD5, CD23, MPO, and the other myeloid and monocytic markers tested. These findings were consistent with B-ALL.

The patient was initially treated with three-drug induction per the AALL0932 trial protocol due to average risk at diagnosis. At day 8, cytogenetic results were reported and remarkable for a severely hypodiploid clone, 28XY, +10, +14, +18, +21 with loss of 4, 9, 11, 12, 17, 19, and 22. The UCSF500 Cancer Gene Panel test (nearly 500 different genes, including the majority of known cancer genes) revealed a neurofibromatosis type I (NF-1) mutation, very commonly seen with hypodiploid ALL. Together with cytogenetic and molecular findings, the final diagnosis was B-ALL with near haploid cytogenetic findings. Due to the presence of this high-risk clone, he was switched to a four-drug induction as per AALL1131. On day 29 at the end of induction (EOI), minimal residual disease (MRD) was 0.009%, which was considered negative according to the Children’s Oncology Group (COG) [6] threshold of 0.01%. The patient tolerated induction chemotherapy well. Consolidation and delayed consolidation were carried out per the AALL1131 control arm with high-dose intrathecal methotrexate used for interim maintenance. The patient was transitioned to maintenance therapy per AALL1131, with an additional block of blinatumomab as per AALL1731 due to his high-risk disease. At the time of writing, there is no evidence of recurrent disease.

## 3. Discussion

The differential diagnosis for a rapidly growing orbital mass in a child includes infectious etiologies, inflammatory conditions (such as orbital pseudotumor), and other childhood malignancies (including rhabdomyosarcoma, neuroblastoma, and lymphoproliferative tumors) [7]. Leukemic orbital masses have been widely reported in patients with AML (known as granulocytic sarcoma, myeloid sarcoma, or chloroma), chronic myeloid leukemia, and other myeloproliferative disorders. However, rarely do orbital masses present in the setting of ALL [8]. This case report is unique in the presentation of the orbital mass prior to any detectable systemic manifestations or laboratory abnormalities.

The actual incidence of orbital involvement in ALL is unknown, and the only consensus is that it is a rare extramedullary manifestation of ALL. Our literature search revealed seven reported cases of hematogenous masses at the initial presentation of childhood ALLs, ranging from 8 months to 11 years of age [8,9,10,11,12,13,14]. Unlike our case, these reports had systemic symptoms such as intermittent fevers, lymphadenopathy, or rhinorrhea, or abnormal CBC at presentation (Table 1). To the best of our knowledge, our case is the first reported case in which an orbital mass preceded detectable systemic evidence of ALL.

ALL is a heterogenous disease both clinically and genetically [2]. Based on immunophenotyping of the leukemic cells, ALL is classified into two main subtypes: B cell lineage (B-ALL) or, less commonly, T cell lineage (T-ALL). B-ALL is the most common ALL subtype and among children, B cell lineage ALL constitutes approximately 88 percent of cases (all seven previously reported cases (and our case) were B-ALL). A retrospective study from Italy included orbital and ocular manifestations of acute childhood leukemia of 180 patients. In both the AML and ALL groups, the presence of specific orbital or ocular lesions was associated with a higher frequency of bone marrow relapses and CNS involvement (*p* < 0.05), leading to a lower survival rate [4]. However, interestingly, a previous case report summarized that all but one of the reported cases of B-ALL with an orbital mass at presentation had undergone remission of their leukemia after chemotherapy with or without radiation, as did our case [11]. Bidar et al. [15] reported five male patients with ALL. Two patients died two and seven years after the finding of the orbital mass, whereas two patients had survival for 11 and 13 years, respectively, at the time of reporting. 

Hypodiploidy (<44 chromosomes) occurs in 2 to 3% of children with B-ALL and is a strong negative prognostic factor [1,16]. Our patient is remarkable for a severely hypodiploid clone, 28XY, +10, +14, +18, +21 with loss of 4, 9, 11, 12, 17, 19, and 22, which is defined as near haploid. 

## 4. Conclusions

This patient highlights the importance of differentiating benign causes of eyelid swelling from more serious causes. The presence of a palpable mass along the rim, as opposed to one at the eyelid margin, argued against a benign lesion such as a chalazion. Furthermore, the presence of “orbital signs”, such as displacement of the globe, restriction of motility, proptosis, or decreased vision are additional red flags. Emergent consultation of an ophthalmologist is necessary to facilitate workup and biopsy of the mass lesion. Interestingly, the pre-operative complete blood count with differential identified no blasts or leukocytosis in this patient. Only a repeat test one week after biopsy was able to detect smears upon manual review undertaken after the orbital tissue diagnosis had been made. Thus, the early tissue diagnosis was key to obtaining a systemic diagnosis before frank systemic manifestations were detectable. 

## Figures and Tables

**Figure 1 diagnostics-11-00025-f001:**
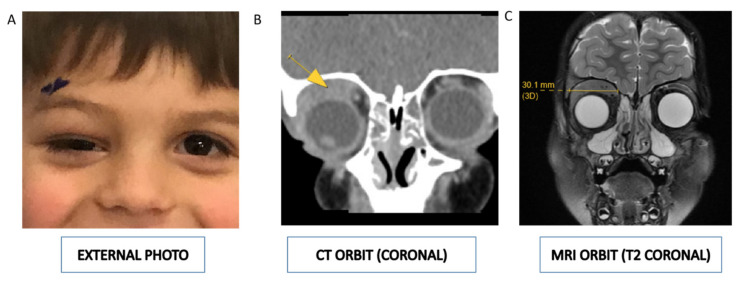
(**A**) External photograph demonstrating fullness of the right upper eyelid with downward displacement of the globe (hypoglobus). (**B**) Contrast-enhanced computed tomography and (**C**) magnetic resonance imaging of the orbit demonstrating a 3 cm by 1 cm mass (indicated by the yellow arrow in Figure 1B) of the right extraconal orbit, suspicious for hematologic malignancy or rhabdomyosarcoma.

**Figure 2 diagnostics-11-00025-f002:**
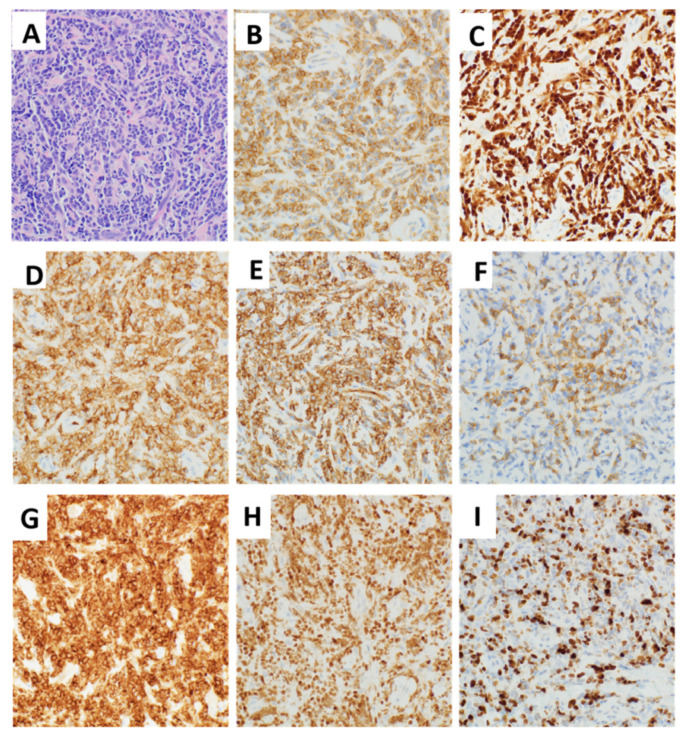
Histologic features of B-lymphoblastic leukemia (B-ALL) from orbital mass. (**A**) Right orbital mass biopsy demonstrating sheets of intermediate-sized immature mononuclear cells with round to irregular nuclear contours, finely dispersed chromatin, inconspicuous nucleoli, and scant cytoplasm (original magnification ×400, hematoxylin and eosin stain). By immunohistochemistry, the cells were respectively positive for (**B**) CD99, (**C**) TdT, (**D**) LCA cocktail, (**E**) CD34, (**F**) CD79a (patchy variable), (**G**) CD10, and (**H**) PAX5. (**I**) Ki-67 was 70–80% (original magnification ×400). All images were taken using an Olympus BX50 microscope and a SPOT Insight 4 camera and SPOT 5.0 advanced software.

**Figure 3 diagnostics-11-00025-f003:**
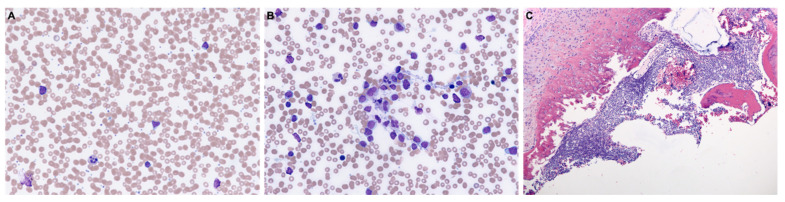
Morphology of B-ALL in peripheral blood and bone marrow. (**A**) Peripheral blood smear and (**B**) bone marrow aspirate smear highlighting numerous circulating small to medium-sized blasts with high nuclear to cytoplasmic ratio, irregular nuclei, open chromatin, small nucleoli, and scant cytoplasm (original magnification ×400, Wright–Giemsa stain). (**C**) Bone marrow core biopsy demonstrating diffuse involvement by B-ALL (original magnification ×100, hematoxylin and eosin stain). All images were taken using an Olympus BX50 microscope and a SPOT Insight 4 camera and SPOT 5.0 advanced software.

**Table 1 diagnostics-11-00025-t001:** Clinical characteristics of previous reported ALL cases with orbital involvement at the time of diagnosis [8,9,10,11,12,13,14].

#	Age	Sex	National Origin	Location Orbital Mass	Laterality	Presenting Signs or Symptoms	Extraorbital Disease at Presentation	Diagnostic Tests Done
1	21mo	M	Iran	Superolateral	L	Proptosis, tenderness, and tearing	Bilateral submandibular lymphadenopathy; abnormal CBC	Bone marrow aspiration
2	3yrs	M	UK	Superior	L	Intermittent lid swelling; mild proptosis and hypoglobus; fatigue, fever, and weight loss.	Multiple lymph nodes; abnormal CBC	Peripheral blood smear and the bone marrow biopsy
3	4yrs	F	India	Superomedial	L	Fevers, bone pain, and proptosis	Generalized lymphadenopathy; hepatosplenomegaly; abnormal CBC	Peripheral blood smear and the bone marrow biopsy; multiplex polymerase chain reactions for t(9;22), t(4;11), t(12;21), and t(1;19)
4	8mo	F	USA	Superior	R	Eyelid swelling and epiphora	Cervical lymph node	Biopsy of the cervical lymph node; bone marrow biopsy; cytogenetic analysis
5	18mo	M	USA	Nasal	L	Ptosis, proptosis, and extropia	Abnormal CBC	Orbital mass biopsy
6	11yrs	F	UK	Lateral	B	Headache; weight loss; painful groin swelling	Inguinal lymphadenopathy	Bone marrow aspiration and trephine; cytogenetics and flow cytometry; biopsy of inguinal node; lumbar puncture
7	3yrs	F	Thailand	Lateral	R	Proptosis; ocular motility deficits	Hepatosplenomegaly; abnormal CBC	Bone marrow aspiration; immunophenotype analysis by flow cytometry; bone marrow cytogenetic study
8	4yrs	M	USA	Superior	R	Eyelid swelling; hypoglobus; ocular motility deficit	None	Orbital mass biopsy; peripheral blood smears; bone marrow aspirate smears; flow cytometry; UCSF500 molecular test (next generation sequencing)

#, patient number in series; F, female; M, male; P, Persian; C, Caucasian; A, Asian; R, right; L, left; B, bilateral; mo, months; yrs, years; CBC, complete blood count.

## Data Availability

Data is contained within the article.

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
