# Peer review of "Hypodiploid B-Lymphoblastic Leukemia Presenting as an Isolated Orbital Mass Prior to Systemic Involvement: A Case Report and Review of the Literature"

_diagnostics, 2020, doi:10.3390/diagnostics11010025_

Round 1

Reviewer 1 Report

The authors present an interesting case of ALL of the orbit without any systemic symptoms.  Although ALL presenting in the orbit is not terribly uncommon, the fact that this patient had this as the initial presentation without any systemic symptoms is interesting.  The study highlights the importance of fully evaluating any patient that presents with periorbital edema, especially with associated orbital signs.

Author Response

Response to Reviewer 1

We thank the reviewer for their time and comments.

Reviewer 2 Report

This is a concise report of an ALL presenting with an orbital mass with no other symptoms or signs. The aurthors provide an excellent review of the literature. 

My main critiques are very minor and as follows:

First 2 sentences in abstract seem out of place. Maybe start with sentence 3, "We describe..." 

"Orbital biopsy consisted with B cell lymphoma..." may be a typo. Do the authors mean "Orbital biopsy was consistent with..."

The sentence "However, by that time..." could perhaps be blended with opening statement. Maybe something like: "We describe a 4-year-old boy who presented with progressive right periorbital edema and proptosis, with no systemic symptoms, found to have pre-B-cell ALL." 

Some other minor grammatical edits are need to these phrases:

"after several chemotherapies" (maybe just "after chemotherapy")

"Previous seven reported" (maybe "There are seven prior cases reported..."

Author Response

Reviewer 2:

This is a concise report of an ALL presenting with an orbital mass with no other symptoms or signs. The aurthors provide an excellent review of the literature. 

My main critiques are very minor and as follows:

1. First 2 sentences in abstract seem out of place. Maybe start with sentence 3, "We describe..." 

Response to Reviewer 2: Thank you for your advice. We have deleted the first two sentences in abstract.

2."Orbital biopsy consisted with B cell lymphoma..." may be a typo. Do the authors mean "Orbital biopsy was consistent with..."

Response to Reviewer 2: We apologize for this mistake and we have corrected it.

3.The sentence "However, by that time..." could perhaps be blended with opening statement. Maybe something like: "We describe a 4-year-old boy who presented with progressive right periorbital edema and proptosis, with no systemic symptoms, found to have pre-B-cell ALL." 

Response to Reviewer 2: The sentence “However, by that time...” was deleted and blended it with opening statement.

4.Some other minor grammatical edits are need to these phrases:

"after several chemotherapies" (maybe just "after chemotherapy")

"Previous seven reported" (maybe "There are seven prior cases reported..."

Response to Reviewer 2: We thank the reviewer for their comment. We have improved the wording as advised.

Reviewer 3 Report

The manuscript was prepared very well. I congratulate the authors for the preparation of the manuscript

However, I have the following comments:

Introduction

Line 39.

You have inserted reference 23. This reference is not in the list.

Lines 39-41.

Could expand the content of the reports of orbital involvement in ALL?

Case Presentation

It is one of the strengths of the article, however the legends of the figures could be improved, for a description that is better understood.

Discussion

Lines 127-133

what are the symptoms of your case? The remaining 7 (6, 12) cases you describe, what diagnostic confirmation tests did you use? It would be interesting to include a table to explain in a simple way what this manuscript provides with respect to the previous cases (6, 12)

Additional

With respect to the methodology used, it is an innovation for this type of case? Could it be standardized for other types of extamedullary locations of ALL? Comment briefly

It makes a brief description of the strengths of its methodology.

Author Response

Reviewer 3:

The manuscript was prepared very well. I congratulate the authors for the preparation of the manuscript

However, I have the following comments:

1. Introduction

Line 39.

You have inserted reference 23. This reference is not in the list.

Response to Reviewer 3: We thank the reviewer for their comment. We regret that this is an automated formatting of references 2 and 3. We hope that this will be addressed in final print, but we will discuss with the editorial staff should this manuscript proceed to proofing.

2. Lines 39-41.

Could expand the content of the reports of orbital involvement in ALL?

Response to Reviewer 3: We appreciate the reviewer’s comment. We have added a large cohort of 288 patients with leukemia to demonstrate the orbital involvement rate in ALL at any time after diagnosis and discussed its prognostic implications.

3. Case Presentation

It is one of the strengths of the article, however the legends of the figures could be improved, for a description that is better understood.

Response to Reviewer 3: We thank the reviewer for their comments. We have edited the legends to improve clarity.

4.Discussion

Lines 127-133

what are the symptoms of your case? The remaining 7 (6, 12) cases you describe, what diagnostic confirmation tests did you use? It would be interesting to include a table to explain in a simple way what this manuscript provides with respect to the previous cases (6, 12)

Response to Reviewer 3: We thank the reviewer for their comment. We have included a new table indicating the symptomatology and diagnostic testing for our case and the previous seven.

5. Additional

With respect to the methodology used, it is an innovation for this type of case? Could it be standardized for other types of extamedullary locations of ALL? Comment briefly

It makes a brief description of the strengths of its methodology.

Response to Reviewer 3: The use of the UCSF500 molecular test (next generation sequencing) in this case is perhaps unique in that most orbital biopsies do not get such molecular sequencing. However, such molecular testing is relatively common in B-ALL so this testing is not necessarily considered an innovation for this type of case. Again it would be if this were a standard pipeline for all orbital masses (malignant or non-malignant) to help with diagnoses. But in the context of pediatric leuekmias/lymphomas, most patients have significant molecular studies performed. The cytogenetic studies were standard as were the radiologic studies.